# Phosphatidyl Ethanolamine Binding Protein *FLOWERING LOCUS T-like 12* (*OsFTL12*) Regulates the Rice Heading Date under Different Day-Length Conditions

**DOI:** 10.3390/ijms25031449

**Published:** 2024-01-24

**Authors:** Yongxiang Huang, Jianfu Guo, Dayuan Sun, Zhenhua Guo, Zihao Zheng, Ping Wang, Yanbin Hong, Hao Liu

**Affiliations:** 1College of Coastal Agricultural Sciences, Guangdong Ocean University, Zhanjiang 524088, China; gdouhrb@126.com (Y.H.); gjf6292@126.com (J.G.); 2Plant Protection Research Institute, Guangdong Academy of Agricultural Sciences, Guangzhou 510640, China; sundayuan@gdaas.cn; 3Rice Research Institute, Heilongjiang Academy of Agricultural Sciences, Jiamusi 154026, China; hljsdsgzh@163.com; 4Department of Agronomy, Iowa State University, Ames, IA 50011-1051, USA; zhzheng@iastate.edu; 5Biotechnology and Nuclear Technology Research Institute, Sichuan Academy of Agriculture Sciences, Chengdu 610066, China; wangping2014@163.com; 6Guangdong Provincial Key Laboratory of Crop Genetic Improvement, Crops Research Institute, Guangdong Academy of Agricultural Sciences, Guangzhou 510640, China; hongyanbin@gdaas.cn

**Keywords:** AT-hook protein, flowering time, *FTL* homologous, H3 acetylation, transcription regulation

## Abstract

Plant *FLOWERING LOCUS T-Like* (*FTL*) genes often redundantly duplicate on chromosomes and functionally diverge to modulate reproductive traits. Rice harbors thirteen *FTL* genes, the functions of which are still not clear, except for the *Hd3a* and *RFT* genes. Here, we identified the molecular detail of *OsFTL12* in rice reproductive stage. *OsFTL12* encoding protein contained PEBP domain and localized into the nucleus, which transcripts specifically expressed in the shoot and leaf blade with high abundance. Further GUS-staining results show the *OsFTL12* promoter activity highly expressed in the leaf and stem. *OsFTL12* knock-out concurrently exhibited early flowering phenotype under the short- and long-day conditions as compared with wild-type and over-expression plants, which independently regulates flowering without an involved *Hd1*/*Hd3a* and *Ehd1*/*RFT* pathway. Further, an AT-hook protein *OsATH1* was identified to act as upstream regulator of *OsFTL12*, as the knock-out *OsATH1* elevated the *OsFTL12* expression by modifying Histone H3 acetylation abundance. According to the dissection of *OsFTL12* molecular functions, our study expanded the roles intellectual function of *OsFTL12* in the mediating of a rice heading date.

## 1. Introduction

Reproduction is a common behavior in organisms belonging to the flowering plants (angiosperms). The process of initial transition to reproduction depends upon sufficient energy accumulation during the vegetative stage by the time the plant is moving into the flowering period [1]. When flowering, the integration and preparation of the plant is a critical precondition for subsequent seed generation, but inception of floral formation is also determined by diverse exogenous environmental factors and endogenous molecular cues. These cues guarantee the transition of the plant into the floral period is coordinated with inner molecular controls, and that the timing occurs in situ with favorable environmental factors to achieve the plants reproductive mission [2]. In *Arabidopsis*, multiple impressive descriptions have unveiled the Darwinian puzzle of plant flowering from its classical genetics all the way to the level of epigenetics [3]. One of the main genes that has been extensively investigated is the *FLOWERING LOCUS T* (*FT*), which is one of the most fascinating discoveries in this field [4]. The *FT* transcript is primarily expressed in the leaves and the encoded protein possesses a conserved phosphatidyl ethanolamine binding domain (PEBP), which mainly interacts with *bZIP* factor *FLOWERING LOCUS D* [5] to promote the transcription of florigen meristem identity genes under long-day conditions and accelerates the converting of the foliar bud into a blossom bud in the shoot apical meristem (SAM). Additionally, *FT* together with *LEAFY* [6], promotes floral meristem development and has an antagonistic relationship with its homologous gene *TFL1* (*TERMINAL FLOWER 1*) [7]. Once FT interacts with TSF (TWIN SISTER OF FT), it inhibits *TFL1* in its determination of inflorescence meristem identity [8]. Despite the aforementioned studies, the field is constantly expanding, and our comprehensive knowledge of flowering behavior is ever increasing; however, functional dissection of the family of *FLOWERING LOCUS T-Like* (*FTL*) genes is still not complete, albeit they are expected to have similar effects to those of *FT* [9].

Rice contains two sub-species, *Xian* and *Geng*, which are widely cultivated in a wide range of global areas and provides basic carbohydrate intake for billions of people. Currently, local farmers prefer to cultivate an early-flowering rice variety that improves the time it takes to induce flowering and is economically efficient; this illustrates that the control of the molecular mechanism of rice flowering is beneficial to actual yield production and modulation of this mechanism will aid in the requirement of the food source for billions. Indeed, rice flowering is coordinated by two photoperiodic pathways controlled by different circadian rhythm conditions. In the short day period, the *Heading data 3a* (*Hd3a*) is homologous with *FT*, and its transcription is activated by *Heading data 1* (*Hd1*), similar to the way in which FT is under the control of *CONSTANS* [10]. Furthermore, *RICE FLOWERING LOCUS T1* (*RFT1*), the closest homolog to *Hd3a*, dominants the floral transition period [11]. However, *RFT* has positive (MADS-box protein gene, *OsMADS50* and *Ehd1*) and negative (*Hd1*, *Phytochrome B phyB* and *Grains Height Date-7 Ghd7*) regulators that form a gene network in order to modulate long-day flowering [12]. During long-day flowering, a rice-specific floral inducer, *Ehd1* (*Early heading data 1*), integrates multiple pathways to mediate *RFT1*, which leads to flowering under the appropriate photoperiod conditions [13]. Although the significant functional diversity of rice flowering has been described by existing studies of *Hd3a* and *RFT*, it is still necessary to interrogate the new available gene, which should be taken into account when further breeding for food production is undertaken.

The *FT-Like* (*FTL*) family of genes is broadly expanded by gene duplication and independently exists in nearly all modern angiosperm species [14]. The emergence of *FTL* genes coincided with the evolution of flowering plants, and the potential ability of these genes to promote flower transition appears to be conserved in angiosperms [15]. *FTL* genes will probably contribute to the next breakthrough in flowering botany research, and recent demonstrations have supported this speculation. For example, *Arabidopsis TSF* encodes a floral inducer homologous with FT, playing overlapping roles in the promotion of flowering [16]. Furthermore, in cereal monocots, wheat *VRN3* (*VERNALIZATION 3*) is the candidate gene for *FT* [17] and facilitates spikelet efflorescence during long-day conditions. Additionally, *HvFT3* accelerates the initiation of spikelet primordia and the early reproductive development of spring barley independently of the photoperiod [18]. In tomatoes, the day-neutral flowering attribute phenotype is controlled by variations in the two *FT* paralogs, *FTL1* and *SELF-PRUNING 5G* (*SP5G*) [19]. Apart from in-flowering regulation, *Arabidopsis MFT* (*MOTHER OF FT AND TFL1*) was reported to respond to seed germination through a negative feedback loop that modulated ABA signaling [20], and finally, rice *OsMFT2* is heavily involved in abscisic acid signaling mediated seed germination through cooperating with multiple *ZIP* transcription factors [21]. In all, the current data suggests that *FTL* genes display versatility in the regulation of plant flowering and other biological functions of plant species.

There are fifteen *FT-like* gens in the rice genome, including thirteen *FTL* and two *MFT* genes; the majority have roles in the process of controlling the induction of the alteration of the plant behavior from the nutritive growth period into the reproductive stage, which have not been fully investigated. Previously, the expression levels of *OsFTL12* and *BBX* (*B-box transcription factor*) genes, and several *CONSTANS-Like* (*COL*) genes were identified as being up-regulated in plants with an AT-hook protein *OsATH1* (*LOC_Os07g08710*) knock-out plants [22]. These plants presented with a delay flowering phenotype. However, the effects of *OsFTL12* on floral transition has not yet been examined. In this study, we identified, in detail, the function of *OsFTL12* during the reproductive development of rice. The results herein expand our understanding of *FTL* genes functions in the regulation of a rice heading date.

## 2. Results

### 2.1. Isolation of OsFTL12 from Rice Genome

The *OsFTL12* gene was isolated from the rice genome sequence based on the gene accession number *LOC_Os06g35940*. *OsFTL12* is located on chromosome 6, the DNA sequence was 2608 bp in length and included four exons and two introns (Figure 1A), the nucleotide length of *OsFTL12* CDS was 522bp (Appendix A), which encodes a protein of 173 amino acid residues with a predicted molecular weight of 19.456 kDa. The NCBI Blast Protein database predicted that the PEBP domain existed in the region of 4–172 in the amino acid sequence (Figure 1B), and the SWISS-model database predicted that the *OsFTL12* protein crystal structure was similar to that of the template of *Arabidopsis FLOWERING LOCUS T* protein (PDB accession 6IGJ) [23] (Figure 1C). To understand the phylogenetic relationship between the *OsFTL12* and other *OsFTL* family genes, 15 genes paralogous to *FT* were obtained from the rice genome database. Phylogenetic analysis suggested that the rice *FTL* family of genes can be divided into three large subgroups, *OsFTL12*, *OsFTL11*, *OsFTL5* and *OsFTL6*, which were divided into a secondary branch in subgroup III (Figure 1D), implying that they probably maintained a similar function in the regulation of rice flowering. Subsequently we searched for homologues of *OsFTL12* in a diverse number of plant species to rebuild the phylogenetic tree. As result, 61 orthologues were obtained from 38 species (Appendix A), of which 24 orthologues and 12 paralogues were used to reconstruct the genetic tree (Appendix A), which indicated that *OsFTL12* was similar to the *BRADI_2g49795v3* gene in *Brachypodium distachyon*. In conclusion, the sequence alignment provided a basic reference for understanding the puntative function of *OsFTL12* in the rice genome.

### 2.2. Subcellular Localization Identified That the OsFTL12 Protein Is Targeted to the Nucleus

The subcellular localization of the transcription factor *OsFTL12* was predicted to be the nucleus by web-tool Cell-PLoc [24]. Next, in order to identify the true targeted subcellular position of the *OsFTL12* protein, *OsFTL12*-GFP and the nucleus markers *OsCOL9*-RFP [25] and *AtCO*-RFP were co-transformed into the protoplast cells of monocotyledon rice and dicotyledon *Arabidopsis*. Consequently, the *OsFTL12*-GFP released green, fluorescent light that merged with the nucleus markers *OsCOL9*-RFP, which released obvious yellow light in the rice nucleus (Figure 2A). The same result was obtained in *Arabidopsis* protoplast cells, whereby *OsFTL12*-GFP overlapped with *AtCO*-RFP red fluorescent (Figure 2B). In both cases, the two light-omitting regions were superimposed to generate an obvious yellow light region in the nucleus. Hence, we concluded that *OsFTL12* was mainly localized in the nucleus.

### 2.3. OsFTL12 Promoter Activity Analysis

Firstly, we isolated the 2 kb promoter sequence of *OsFTL12* from the rice genome library (Appendix A) and the plant *cis*-acting regulatory DNA elements (PLACE) [26] were carried out to predict conversed *cis*-elements in the *OsFTL12* promoter. A total of 67 and 58 conserved binding motifs were predicted to be found in the positive (+) and negative (−) DNA strands of *OsFTL12* the promoter, respectively. A total of 38 CAAT-box binding motifs were predicted to exist in *OsFTL12* promoter, which would be expected in a common *cis*-acting element often found in promoter and enhancer regions. The number of core transcription start elements, TATA-boxes and AT~TATA-boxes, ranked second only to the total number of *cis*-elements. In addition, multiple motifs were involved in responding to plant hormones that were also found in the *OsFTL12* promoter region, including four-JA, two-ABA, and single occurrences of -SA, -IAA and-GA motifs. These data implied that *OsFTL12* may be activated in response to phytohormone signaling (Appendix A). Furthermore, analyses of the PlantTFDB database [27] and Plant regulomics database [28] were carried out to predict the TFs upstream of *OsFTL12* in rice and the *Arabidopsis* genome. In doing so, it was identified that at least 104 rice (Appendix A) and 69 *Arabidopsis* (Appendix A and Appendix A) TFs were predicted to bind 78 and 215 conserved motifs in *OsFTL12* promoter, respectively. Of these, *ethylene response factors* (*ERF*), *MYB* and *WRKY* transcription factors ranked in the top three most occurrent groups. Overall, the prediction of the *cis*-elements and transcription factors for *OsFTL1* indirectly provided a useful tool for further designing experiments to understand the involvement of *OsFTL12* in the biophysiological reaction pathway of plants.

To identify the mode of expression of *OsFTL12* in different tissues, the promoter sequence of *OsFTL12* was inserted into a GUS (β-glucuronidase) reporter vector to evaluate activity of the promoter in the P*_OsFTL12_*-GUS transgenic line. The analysis of GUS staining indicated that the *OsFTL12* promoter was weakly activated in shoot epidermis cells but was not activated in cells of the young root or germinated seed shoot tip (Figure 3A). After the P*_OsFTL12_*-GUS transgenic plant grew into the reproductive stage, we utilized the tissues of the matured plant to repeat this experiment. The histochemical staining result suggested that the *OsFTL12* promoter was still not driving the GUS reporter gene expression in cells of the matured root, panicle or the flower. However, it was slightly activated in several tissues, such as the matured leaf blade (Figure 3B) and fresh stem (Figure 3C). Generally, even though we were able to amplify the whole length of the CDS of *OsFTL12* from various tissues, its promoter was characterized by a tissue-specific expression pattern.

### 2.4. Identification of the Expression Pattern of OsFTL12

Furthermore, to understand whether the *OsFTL12* gene has intrinsic tissue specific expression ability, we obtained original TPM (Transcripts Per Million) values of *OsFTL12* transcripts based on 425 RNA-seq samples published in the Rice Expression Database (RED) [29]. We then rebuilt a visual histogram indicating that *OsFTL12* was preferentially highly expressed in cells of the shoot, endosperm, node, callus, and leaf (Figure 4A and Appendix A). This result was consistent with the Electronic Fluorescent Pictograph Datasets previously published [30] (Appendix A).

Furthermore, we detected the truth expression levels of *OsFTL12* in different tissues of the WT plant at multiple stages by quantitative, real-time PCR. The results indicated that *OsFTL12* was amplified in various tissues at different stages, with the highest expression levels in the shoot, followed by the leaf blade, stem, and spikelet. Conversely *OsFTL12* was expressed at low levels in the root, flower, and development seed. The abundance of the *OsFTL12* transcript gradually increased in the leaf during development of the plant from seedling to heading stage, implying that *OsFTL12* is probably involved in the process of rice leaf growth (Figure 4B). These results showed that *OsFTL12* was obviously preferentially expressed in a tissue-dependent manor.

### 2.5. Construction of OsFTL12 Over-Expression and Knock-Down Transgenic Plants

To more conclusively understand the molecular roles of *OsFTL12*, we constructed the following vectors: Ubi-*OsFTL12* (*OsFTL12-OX*), and Ubi-CRISPR/Cas9 bound to U3-*OsFTL2*-gRNA (*osftl12-ko*), both of which were induced into the callus of wild-type plant by utilizing *Agrobacterium*, strain EHA105 (Appendix A). Finally, we obtained six plants of overexpression and real-time PCR associated with a Hygromycin marker was used to examine the expression level of *OsFTL12* in these plants (*OsFTL12-OX*). One independent line, which contained the highest expression level of *OsFTL12*, was identified that had more than a 10-fold increase in T1 generation (Figure 5A), and three other independent plants (T2) were identified and used in the subsequent experiments whereby we screened the offspring population of *OsFTL12-OX* for T1 generation (Figure 5D).

For the generation of a knock-out plant, a specific primer was synthesized, based on the design of a small guide RNA (*OsFTL12*-gRNA), to amplify the DNA fragment in the *OsFTL12* genomic region. After sequencing of the amplified target DNA, we identified three knock-out plants, all of which exhibited DNA-base mutations in the *OsFTL12* sequence when compared with the wild type. Furthermore, to evaluate the potential risk of off-target induction in *osftl12-ko* plants, CRISPR-GE database [31] interrogation was firstly carried out by blasting the rice genome with our designed small gRNA (20 bp), which led to the prediction that 13 genes were potential sites for off-target induction (Appendix A). Additionally, large-scale off-target examination was performed on the *osftl12-k* T1 plants, and one line presented with Cas9-sgRNA*_OsFTL12_*, which caused mismatched splice sites that did not exist in the predicted off-target genes sequences. This was one of the positive transgenic T1 lines of *osftl12-ko* (Figure 5B,C). Finally, we determined three independent T2 lines by screening the offspring of the *osftl12-ko* T1 generation (Figure 5E). Together, the stably expressed seeds of the T2 generation were successfully harvested from the *OsFTL12* functional knock-out and knock-in transgenic plants for use in later experiments.

### 2.6. OsFTL12 Over-Expression Delays Heading Date in Rice

When we obtained the *OsFTL12* transgenic lines, we began to investigate the differences in the flowering time of the *OsFTL12* transgenic and wild-type plants when grown in SD and LD conditions. The average heading time (the time in which it took the seed germination to reach flowering) of the wild-type plants were 65 days (SD) and 68 days (LD) (Figure 6A,B). Two *osftl12-ko* lines heading date were 57–58 (SD) and 60–61 days (LD), respectively. The phenotype of *OsFTL12* knock-out plants was compared to the wild-type plants, which began to flower one week earlier under both LD and SD day-light conditions. Meanwhile, two over-expressions of *OsFTL12* (*OsFTL12-OX*) lines flowering time were 70–71 days (SD) and 73–76 days (LD), the phenotype of heading time investigation in *OsFLT12-OX* lines remained at the vegetative growth stage after the flowering of the wild-type and *osftl12-ko*. Especially during the LD day-light time conditions, overexpression of *OsFTL12* delayed the flowering time about one week later than the wild-type plant. Thereby the *OsFTL12* mutant displayed a promoted reproductive stage (Figure 6C,D).

Previous reports have indicated that rice flowering is controlled by a photoperiodic triggered circadian rhythm reaction [32]. To understand the circadian rhythm expression pattern of *OsFTL12* under different light conditions, we extracted the total leaf blade mRNA from wild-type plants, and then detected the *OsFTL12* expression level using qPCR at 4 h intervals. Under SD conditions, the *OsFTL12* expression level was activated during the day-light and was gradually elevated from 8:00 a.m. till 12:00 p.m., but was then slowly down regulated until 8:00 a.m., before being elevated again at 8:00 p.m. until 12:00 a.m. Generally speaking, *OsFTL12* transcription was maintained at a low level, and did not display obvious fluctuation (Figure 6E). Under LD conditions, *OsFTL12* expression levels in the daytime were higher than those at nighttime, where it peaked at 12 a.m., and then gradually decreased until about 8:00 p.m. (Figure 6F). In conclusion, the detection of circadian pattern suggests that the up-regulation of expression of *OsFTL12* was activated by light and had a tendency to vary under different the day-light conditions. There was no obvious change with the extension of day-light time.

### 2.7. RNA-Seq Identified the Differentially Expressed Genes (DEGs) in osftl12-ko Plants

To better understand the downstream gene network in *OsFTL12* deficient plants during the rice reproductive period, RNA-seq was used to investigate the differentially expressed genes (DEGs) in WT and *osftl12-ko* lines at the booting stage. Firstly, the majority mapped ratio of each library matched to the existing rice genomes at more than 95% (Appendix A); principal component analysis (PCA) clearly divided the samples from WT and *osftl12-ko* (Appendix A), implying the RNA-seq data had a high degree of confidence for a next analysis. A total of 1629 annotated DEGs and 65 new transcripts were identified in the *osftl12-ko* profile, of which 1169 up-regulated and 525 down-regulated DEGs had the cut-off value > 1-fold change (Appendix A and Appendix A). A further 322 and 114 DEGs were identified as being increased/down-regulate with a 2-fold (Log_2_FC > 2) change in *osftl12-ko* profile, by improving the filtering parameter (Appendix A). However, the genes related to the family of *B-BOX*, *COL* and *TOC1*, which are involved in the molecular function of flower regulation, were not shown to be changed in *osftl12-ko* plants compared to WT. Next, the attributes of all the identified DEGs were classified via Gene Ontology (GO) categories and KEGG enrichment. In terms of molecular function, the majority of the DEGs encoded proteins attributed to three groups including binding, catalytic activity, and transporter activity (Appendix A). KEGG pathway enrichment analysis suggested that *OsFTL12* deficiency mainly affected starch and sucrose metabolism, protein processing in the endoplasmic reticulum, and plant hormone signaling transduction pathway (Appendix A); this implies that *OsFTL12* probably responds to ER stress and metabolism issues in the rice plant.

### 2.8. OsFTL12 Independently Regulate Rice Early Flowering

To validate whether the circadian rhythm variation in regulation of rice flowering main regulators, *Hd1*/*Hd3a* and *Ehd1*/*RFT*, was modulated by *OsFTL12* mutation, the expression patterns of the two genes were detected in wild-type and *OsFTL12* transgenic plants under different daylight modes. The expression of the transcripts of the main flowering regulators, *Hd1* and *EHd1,* were maintained at a relative stable level in *OsFTL12-OX* and *osftl12-ko* plants; However, *Hd1* expression levels were clearly down-regulated during daylight under a long-day condition, with changes in the expression of both *Ehd1* and *Hd1* presenting a similar pattern in variation. The expression of both genes was similarly repressed at the time-point during which the light energy accumulated to a higher level. Additionally, *Hd3a* and *RFT* expression levels were slightly up-regulated in *OsFTL12-OX* lines, and their trends in variation showed a similar pattern (Figure 7A). Overall, *Hd1*/*Hd3a* and *Ehd1*/*RFT* followed a similar expression pattern in both wild-type and *OsFTL12* transgenic lines, implying that *OsFTL12* may act as an independent regulator of *Hd1*/*Hd3a* and *Ehd1*/*RFT* to promote flowering. Additionally, according to the RNA-seq result, we further analyzed the transcript abundances of *Hd1*/*Hd3a* and *Ehd1*/*RFT*, this result suggested that the *osftl12* mutant up-regulated the *Hd1* and down-regulated the *Hd3a*, *Ehd1*, and *RFT*. However, the transcript abundances of *Hd1*, *Hd3a*, *Ehd1* showed no significant difference (*p*-value > 0.01) between the wild-type plant and *osftl12-ko*, and the wild-type *RFT* transcript abundance presented not increasing of 2-fold changes to compare with *osftl12-ko* (Figure 7B). Our study displayed that *OsFTL12* did not significantly affect the expression levels of *Hd1*/*Hd3a* and *Ehd1*/*RFT*, which independently modulated the rice flowering time under different light conditions.

### 2.9. Knock-Out OsATH1 Enhances the H3 Acetylation Level in OsFTL12 Promoter Region

Previously, an AT-hook containing DNA-binding protein *OsATH1* (*LOC_Os07g08710*) [22] was discovered that was capable of acting as an upstream factor to regulate the expression levels of *OsFTL12*, by analyzing the transcriptome profile in *osath1* mutants (Figure 8A). In the *OsATH1* knockout line, the expression level of *OsFTL12* was found to be up-regulated at least 2-fold (Figure 8B). Even though *OsATH1* mutation promoted an increase in *OsFTL12*, the mechanism needed to be elucidated. Previous study has shown the AT-hook TFs epigenetically regulate chromatin remodeling and Histone H3 acetylation [33]; it was deduced that *OsATH1* possibly modified the acetylation level of H3 to modulate *OsFTL12* transcription. To verify this hypothesis, a ChIP-qPCR with H3ac antibody was preformed to analyze the H3 acetylation levels of the *OsFTL12* promoter in *osath1-ko* plants (Appendix A). The result suggested that H3 acetylation (H3Ac) abundance was increased by nearly 3-fold (BS1 and BS2) at the AT-rich region of the *OsFTL12* promoter in os*ath1-ko* plants compared to wild types (Figure 8C,D). In conclusion, it can be proposed that *OsATH1* mutation activates the transcription of *OsFTL12* by regulating the H3ac levels at the AT-rich region of the *OsFTL12* promoter, thereby increasing the level of *OsFTL12* transcription to delay the rice heading time.

## 3. Discussion

Reproduction begins in most higher plants following the flowering transition, regulation of fertilization is a crucial point in ensuring the survival of offspring through seed generation, particularly under unfavorable environmental conditions [34]. The appearance of flowering plants accompanies a long-term evolution and an ability to fill a multitude of ecological niches, ensuring natural selection spreads the area of plant growth; however, insight into the process of flower transitioning is still lacking [35]. Moreover, flowering plants are photoperiodic and sensitive to climate changes, this allows the plant to integrate external stimuli with complex internal signaling, garnering a greater degree of control over flowering time and maximizing the chance of successful reproduction through sufficient seed production [36,37]. The regulatory mechanism of flowering has initially been centered on the florigenic protein *FLOWERING LOCUS T*; *FT-like* genes are being identified as increasingly important in the control of flowering, with *FTLs* potentially having a supplementary effect on the main flowering executor *FT* [38].

In total, fifteen *FTL* genes in the rice genome all contain the conserved domain of phosphatidyl ethanolamine binding protein (PEBP) which is present in all three major phylogenetic divisions (eukaryotes, bacteria, archaea). PEBP is found in the mammalian *Raf kinase inhibitory protein* (*RKIP*), which inhibits *MAP kinase* (*Raf-MEK-ERK*), *G protein-coupled receptor* (*GPCR*) kinase and *NFkappaB* signaling cascades [39]. Although the overall structure of PEBP proteins is similar, the members of the PEBP family have very different substrates and oligomerization states. Here, we decoded the molecular roles of *OsFTL12* in the rice reproductive stage. *OsFTL12* knock-out lines accelerated the speed of the heading date under SD and LD conditions, in comparison with a wild-type plant (Figure 6A,B); this directly confirms the importance of *FTL* genes in the flowering process. Rather than working as a supplement to the *FT* pathway, the data generated in this suite of experiments suggest *OsFTL12* functions independently of *FT*; the lack of expression change in the flowering regulators *Hd3a* and *RFT* suggest that the rice flowering network is controlled by *FTL*’s independent of *FT*. Meanwhile, RNA-seq revealed that knocking out *OsFTL12* did not affect the transcript abundances of *B-BOX*, *COL*, and *TOC1* family genes that have been reported as key players in the regulation of flowering. KEGG assessment supported the speculation that *OsFTL12* deficiency influences rice grain quality, causing issues in the metabolism of starch, fatty acid, and protein processing in the ER, implying that rice FTL members contain multiple unknown functions. Together, this study expands our understanding of the roles of PEBP genes and provides *OsFTL12* as a potential genetic resource for the next early-heading breeding project in rice.

In addition, the full-length coding sequence of *OsFTL12* was easily amplified from mRNA libraries of different tissues, indicating that *OsFTL12* is expressed in multiple organs at varying levels. However, the GUS staining reporter system of the *OsFTL12* promoter did not show results consistent with real-time PCR; this may be due to the fact that *OsFTL12* promoter activity is weak to promote β-glucuronidase reporter gene expression. The GUS assay suggested that *OsFTL12* is not a highly expressed *FTL* gene with the obvious feature of tissue specific expression (Figure 3). According to the analysis of the *cis*-acting elements in the *OsFTL12* promoter, it was found that the CAAT-box and AT~TATA-box broadly distributed at the region of 0 to −1000 bp are located upstream of 5′-UTR. The CAAT-box is a common element in both the promoter and enhancer region [40]. The TATA-box is core promoter element around −30 bp at the start of the transcript. Both of them possibly play a critical role in the transcription of *OsFTL12*. Currently, many transcription factors have been reported to act as an upstream factor in the regulation of *Hd3a* and *RFT* expression, such as *CONSTANS-Like* genes *Hd1* [41] and *OsCOL4* [42], PPR family gene *DTH7* (*Days to heading 7*) [43], and *GARP DNA-binding* gene *Ehd1* [44]. Indeed, the Plant TFDB database predicted that the majority of TFs that are located upstream of *OsFTL12* are integral for plant growth and the stress response. These are not found to be controlled by typical flowering TFs, which again implies that *OsFTL12* maybe depends upon these TFs to respond to another growth signaling; however, further evidence is needed to back this up. *OsFTL12* deficiency up-regulated the expression of genes found in the plant hormone, signaling pathway and pathways involved in pathogen interaction by RNA-seq, which may be due to *cis*-elements in the *OsFTL12* promoter that respond to hormones.

An ignored truth is that a lot of AT-rich sequences exist in the *OsFTL12* promoter, as demonstrated by PLACE database predictions. AT-rich sequences are initiation regions flanked by more GC-rich regions and located predominantly in intergenic regions, which modulate DNA replication [45]. *OsFTL12* transcription may be involved in the DNA replication process, with the AT-hook protein most likely involved in this interaction. Recently, evidence has proved that AT-rich sequences are generally found in flanking regions, which specifically bind to AT-rich sequences in the nuclear matrix attachment regions (MARs); these activate or suppress the expression of their target genes via epigenetic modification, such as histone acetylation and DNA methylation [46]. In total, 29 AT-hook TFs in the *Arabidopsis* genome, such as *AT-HOOK MOTIF NUCLEAR-LOCALIZED PROTEIN 16* (*AHL16*), protect the stability of the genome stability by recruiting *FVE* (*AT2G19520*) and jointed *MSI5* (*AT4G29730*). These contain *HDAC* (*Histone Deacetylases*) complexes that target different loci including *FLOWERING LOCUS C* (*FLC*), *FLOWERING WAGENINGEN* (*FWA*), and *TETRASPORE kinesins* (*TEs*); this promotes a cycle of histone deacetylation, which induces transcriptional silencing [47]. Indeed, overexpression of *AHL22* inhibits flowering by modifying histone H3 acetylation levels in the *FT* region. Although AT-hook TFs have been dissected in the regulation of *Arabidopsis* flowering, they have not been investigated in rice flowering. The AT-hook in the *OsATH1* mutant caused an increase in *OsFTL12* expression, through the up-regulation of histone H3 acetylation levels at the AT-rich region (Figure 8E). This result indirectly indicated *OsFTL12* is involved in rice flowering behavior maybe through epigenetic control [48]. Future experiments intend to investigate the binding relationship between the *OsATH1* and Os*FTL12* promoter, and explore the mechanism of *OsATH1* epigenetically control in rice flowering.

Even though *OsFTL12* contained the highest expression levels in rice stem and leaf by GUS-staining and tissue qPCR analysis, *OsFTL12* knock-out not significantly affected the stem or leaf growth. It is known that the stem and growth depend on the phytohormone pathway, especially the gibberellin (GA), brassinolide (BR), and strigolactone (SL) pathways by modulating the critical genes expression, such as the *SD1*, *OsDWARF4*, *D14*. However, we found that *osftl12-ko* up-regulated the expression levels of five *GA 2-beta-dioxygenase genes* (*LOC_Os07g01340*, *LOC_Os01g55240*, *LOC_Os04g44150*, *LOC_Os02g41954*, *LOC_Os05g43880*), and down-regulated one *GA 20 oxidase gene* (*LOC_Os03g63970*), and the transcriptome DEGs profile did not contain the BR and SL related genes. Even though the *OsFTL12* affected the expression level of GA biosynthesis genes, and did not produce an effect on the BR and SL pathway. We deduced that antagonism procedure existed in *OsFTL12*-induced GA genes, such as the *OsGA2ox9* (*LOC_Os02g41954*) up-regulated reduced the stem length, but down-regulated the *GNP1* (*LOC_Os03g63970*) promoted the GA content increasing; therefore, the GA synthesis genes did not affect the stem or leaf growth. On the other hand, *OsFTL12* expression levels shown higher in stem and leaf, did not represent that the *OsFTL12* protein content had the similar tendency to regulate the stem and leaf growth. Finally, we speculated that *OsFTL12* lost function, maybe another rice *FTL* homologous genes replaced the *OsFTL12* deficient to maintain the normal plant growth, but our speculation still needs to be validated by a next study.

Altogether, flowering or heading is an important agronomic trait used to produce better yields in rice breeding [49]. Conventional improvement methods rely on the field selected, early heading mutants for subsequent breeding. However, in the genome editing era, this can be much improved. The discovery of a novel area of research through the assessment of homologous of *FTL* has led to a break in the over-dependence on *Hd3a* and *RFT* mutants, this allows for the creation of new germplasm resources in the production of early maturing varieties of rice for improving of rice early maturing variety. *OsFTL12* function dissection has proved to be massively beneficial in understanding *FTL* mediated flowering control independent of *FT* and is an area of interesting further research.

## 4. Materials and Methods

### 4.1. Experiment Materials and Growth Conditions

Plant material was taken from the ‘Nipponbare’ cultivar of the rice plant *Oryza Sativa L. japonica*, this was used as the wild-type plant in this study. The ‘Nipponbare’ seed was collected and stored as a rice germplasm by Prof. Jianfu Guo in the College of Coastal Agricultural Sciences, Guangdong Ocean University. Wild-type and transgenic plants were cultured in a constant temperature (28 °C daylight, 25 °C dark) and humidity incubator with 50%. The plants were cultured on short (SD, 10 h daylight, 14 h dark) and long day (LD, 14 h daylight, 10 h dark) light cycles, respectively. Light intensity for plant growth is typically measured as photosynthetic photon flux density (PPFD); the chamber PPFD value was 500 µmol·m^2^·s^−1^, this parameter is eminently suited for plant growth. The seeds of all plant materials germinated from the soil; this day was recorded as the first day. The heading date was recorded as the day when the rice panicle grew out from the leaf sheath with 5 cm, and 30 independent plants were recorded as their heading date. When the different rice materials (wild-type and transgenic plants) flowering rate was 90%, all the plant materials were removed from the chamber to a greenhouse until harvested their seeds [25].

### 4.2. Bioinformatics Analysis

Firstly, 15 homologous’ sequences of the *FLOWERING LOCUS T-Like* were obtained from the rice genome by blasting the rice database (https://www.ricedata.cn). Phylogenetic analysis was performed using the online tool EvolView version 3 [50] by the neighbor joining method. A Phylogenetic tree of *OsFTL12* orthologues in different plant species was generated by blasting the *OsFTL12* sequence in Ensembl Plants database (www.plants.ensembl.org, accessed on 1 December 2020). Gene and protein structures were drawn using IBS 1.0 version software [51]. The NCBI Protein Blast database predicted the crystal structure of phosphatidyl ethanolamine binding protein (PEBP) in the *OsFTL12* protein sequence. SWISS-modeling [52] predicted the 3D crystal structure template for the *OsFTL12* protein. PLACE [26] predicted the cis-elements in the *OsFTL12* promoter. The Plant regulomics database [28] predicted the *OsFTL12* upstream putative binding TFs by blasting rice and *Arabidopsis* genomes. The Rice Expression Database [29] provided 425 RNA-seq transcripts per million values of *OsFTL12* in distinctive tissues.

### 4.3. Subcellular Localization Analysis

The full-length *OsFTL12* CDS without a stop codon was cloned into the *Xba*I/*Bam*HI sites of vector pNA580 (35S:GFP). The resulting 35S-*OsFTL12*-GFP vector and nucleus marker were transiently co-transformed into rice and *Arabidopsis* protoplast cells using polyethylene glycol PEG4000 (CAS#25322-68-3, Sigma-Aldrich, Saint Louis, MO, USA). The nuclear markers were 35S-OsCOL9(*LOC_Os03g50310*)-RFP [25] and 35S-AtCO(*AT5G15840*)-RFP. The full protoplast cell isolation procedure was followed in accordance with a previous published protocol [53]. Briefly, four-week old *Arabidopsis thaliana* seedling leaves (ecotype *Columbia*) and two-week old rice seedling (wild-type plant, *Oryza Sativa L. japonica* cv. Nipponbare) sheaths were cut into approximately 0.1 mm strips. The strips were then incubated in a 10 mL enzyme solution with 2% Cellulase “Onozuka” R-10 (Yakult, Kitakami-shi, Japan), 0.75% Macerozyme R-10 (Yakult, Japan), 0.6 M mannitol (CAS#69-65-8, Merck, Darmstadt, Germany) for 4 h under dark condition. Protoplast cells were mixed with 10 μg of vector and 110 μL 40% PEG4000, the collected protoplasts were cultured at 28 °C for 16 h. Fluorescence was observed by laser scanning using a confocal microscope Zeiss LSM 780 (Carl Zeiss Microscopy GmbH, Jena, Thuringia, Germany).

### 4.4. GUS Assay

The ‘Nipponbare’ was used as the wild-type plant for the transgenic experiment. The callus was induced by IAA (CAS#87-51-4, Macklin, Shanghai, China) and kinetin (CAS#525-79-1, Macklin, China) in the MS medium. Then, the 2 kb promoter sequence of *OsFTL12* was cloned from the wild-type genome and inserted into the *BgI*I/*Bam*HI (New England Biolabs, Ipswich, MA, USA) sites of the pCAMBIA1305 vector. The vector promoter*_OsFTL12_*-GUS was transferred into the Agrobacterium EHA105 to infect the ‘Nipponbare’ callus and generate a transgenic plant. The different tissues of the promoter*_OsFTL12_*-GUS transgenic plants were incubated with the GUS staining kit (Cat.#BL622A, Biosharp, Shanghai, China) for 12 h at 37 °C. Ethyl alcohol (CAS#64-17-5, Macklin, China) was used to de-stain the chlorophyll three times and the sections observed by optical microscope [54].

### 4.5. Total RNA Extraction, Real-Time PCR Analysis of Gene Expression

RNA extraction and qPCR followed our previous published study [25]. For flowering genes expression (*OsFTL12*, *Hd1*, *Hd3a*, *Ehd1*, *RFT*) examination, the rice materials were firstly cultured for 45 days in the chamber under SD and LD condition, the rice leaf blade was harvested at four hour intervals, where mRNA was then extracted to detect the circadian expression patterns of the relevant genes. Total RNA was extracted from 100 mg of 45 day-old wild-type and transgenic (*OsFTl12-OX*, *osftl12-ko*, *osath1-ko*) plants leaf using Trizol Reagent (Cat#15596026, Invitrogen, Carlsbad, CA, USA), then reverse-transcribed using a PrimeScript RT reagent Kit (Cat#FSK-100, TOYOBO, Osaka, Japan). The 100 ng cDNA was quantified in a 20 µL reaction volume with SYBR Premix ExTaq™ (Cat#RR390A, TaKaRa, Osaka, Japan) by using the ABI StepOne Plus system (Cat#4376600, Applied Biosystems, Foster City, CA, USA). The rice actin gene was used as an internal reference. The relative Gene expression levels as a fold change compared to control sample (wild-type) were calculated using the 2^−ΔΔCT^ calculating method with three biological repeats [55]. For *OsFTL12* expression in different tissues following the same method, the total mRNA was extracted from the one week-old wild-type (Nipponbare) seedling root, sheath, leaf; 45 day-old wild-type shoot, stem, mature leaf; 65 day-old wild-type panicle, spikelet, flower; and the seed developed at 5, 15, 25 day during the period of grain-filling.

### 4.6. Generation of the Transgenic Plants

*OsFTL12* cDNA was isolated from the rice seedling DNA library and inserted between the maize ubiquitin promoter and the Nos terminator in the plant over-expression vector, pOX. CRISPR/Cas9 technology was used to generate *osftl12-ko* plants [56]. Briefly, a 20 bp DNA fragment that included a protospacer-adjacent motif (PAM) was designed as a small guide-RNA for *OsFTL12* and fused with a U3-gRNA box. The resulting U3-*OsFTL2*-gRNA fragment was cloned into the *Bsa*I (New England Biolabs, USA) site of pYLCRISPR/Cas9PUbi-H. The *pOX-OsFTL12* and *Cas9-OsFTL12* vectors were introduced into the *Agrobacterium* EHA105 to transform the ‘Nipponbare’ callus. Transgenic rice plants were regenerated from the transformed callus using selection media containing 50 mg·L^−1^ hygromycin. The *OsFTL12* expression levels in the transgenic rice plants were further confirmed with real-time PCR and DNA fragment sequencing. For obtaining the *osath1-ko* line, our study previously had constructed the *osath1-ko* plants in ‘Nipponbare’.

### 4.7. RNA-Seq

Total RNA was extracted from 45 day-old wild-type and *osftl12-ko* leaves (mixed three blades, liquid nitrogen ground 100 mg leaf powder), using the reagent TRIzol (Cat#15596026, Invitrogen, USA). The quality and quantity of each RNA sample were analyzed using NanoDrop 2000c (Thermo Scientific, Waltham, MA, USA) and the Agilent 2100 Bioanalyzer (Agilent, Santa Clara, CA, USA), respectively. Equal quantities of total RNA from each sample were purified and followed by the next library preparation. Total RNA was used for library construction following the High-Throughput Illumina Strand-Specific RNA Sequencing Library protocol. The RNA libraries were sequenced on an Illumina HiSeq2000 instrument (ILLumina, San Diego, CA, USA) to produce 150 bp paired-end reads, each sample (wild-type and *osftl12-ko*) had three independent biological repeats. The last assembled reads were mapped to the Rice Genome Annotation Project (RGAP) database by software Tophat version 2.1.1. A bioinformatics analysis was completed by the company Biomarker Technologies. Briefly, Cufflinks software (version 2.2.1) was used to compute the relative expression of total genes, allowing five mismatches at most. Reads with multiple matches were removed from the primary search results. For each pair of forward and reverse reads, both ends were required to uniquely map to the same transcript. After these filtrations, a set of uniquely mapped pairs were collected for the subsequent abundance estimation. Using the uniquely mapped read pairs, the expression levels of the transcripts were estimated with Fragments Per Kilobase of transcript per million mapped fragments (FPKM). Further FPKM values were calculated by adopting an in-house script, according to the count table of the assembly sequences output. Differentially expressed genes (DEGs) were determined from different samples using an R package with DEGseq analysis protocol. For each gene, the *p*-value and Q-value were calculated. Then, the significant threshold to control the FDR < 0.05 at a given value was computed. To facilitate graphical interpretation of samples relatedness, principal component analysis (PCA) was carried out to detect the major source of expression variances underlying development using R package version 4.3.2. The up and down-regulated DEGs were identified by applying the Cuffdiff module in the Cufflinks software version 2.2.1 with a cutoff value of Log_2_ Fold Change (Log_2_FC) > 1 and <−1 (*p* < 0.05), respectively. Gene ontology analysis of DEGs was performed using GO web-based tools. KEGG pathway enrichment was employed using the online KEGG web server. Original raw sequence data has been uploaded into the National Genomics Data Center (https://ngdc.cncb.ac.cn) Genome Sequence Archive (GSA), and the assigned accession of the submission was CRA003052.

### 4.8. Chromatin Immunoprecipitation (ChIP) qPCR Assay

Two-week old rice seedlings (wild-type and *osath1-ko*) grown under dark conditions, with yellowing leaves, were vacuum infiltrated with 1% formaldehyde for cross-linking and ground in liquid nitrogen after quenching the cross-linking process. Chromatin preparations were sonicated into 0.2–0.5 kb fragments. Specific antibodies against H3Ac (Cat#06-599, Merck, Germany) were added to the chromatin solution, which was precleared with salmon sperm DNA/Protein A agarose beads (CAS#438545-06-3, Sigma-Aldrich, USA). The precipitates were eluted from the beads. Next, the cross-links were reversed, and residual proteins were removed by incubation with proteinase K (CAS#39450-01-6, Merck, Germany). DNA was recovered using a QIAquick spin column (Cat#28104, Qiagen, Valencia, CA, USA) and quantitative PCR was used to determine the amounts of genomic DNA enriched in the chromatin samples. The reference gene *OsUbq13* was used as an internal standard for normalization. The primers were designed previously to amplify DNA fragments of 150–250 bp [57]. The relative gene expression levels as a fold change compared to control sample were calculated using the 2^−ΔΔCT^ calculating method.

### 4.9. Statistical Analysis

*OsFTL12* expression level in different tissue was calculated using the real-time PCR value with three independent repeats, the histogram represented the mean ± SD, and significance analysis was carried out with a *t*-test (* *p*-value < 0.05, ** *p*-value < 0.01). *OsFTL12* expression examination in transgenic plants followed the above mentioned statistical method. The plant materials heading date was recorded as the day when the panicle grew out from the leaf sheath with 5 cm. In total, 30 independent plants (n = 30) were calculated for the flowering time, the histogram represented the mean ± SD, and significance analysis was carried out with a *t*-test (* *p*-value < 0.05, ** *p*-value < 0.01). Moreover, rice leaf blade was harvested at four hour intervals and extracted the total mRNA to examine the expression levels of *Hd1*, *Hd3a*, *RFT*, *Ehd1* under SD and LD, the line chart indicated their expression levels with mean ± SD in three independent repeats. All the statistical analysis figures were employed using the GraphPad Prism 9.

## 5. Conclusions

*OsFTL12* encodes a nucleus localized protein with a conserved PEBP domain, which highly expresses in the shoot and leaf and over-expression delays the rice flowering time under SD and LD growth condition. Additionally, *OsATH1* acts as a transcription factor at the upstream of *OsFTL12* promoter to regulate the *OsFTL12* expression by H3 acetylation pathway. Our study contributes to a theoretical basis for future breeding the early-maturing variety by manipulating the expression of *OsFTL12* in rice.

## Figures and Tables

**Figure 1 ijms-25-01449-f001:**
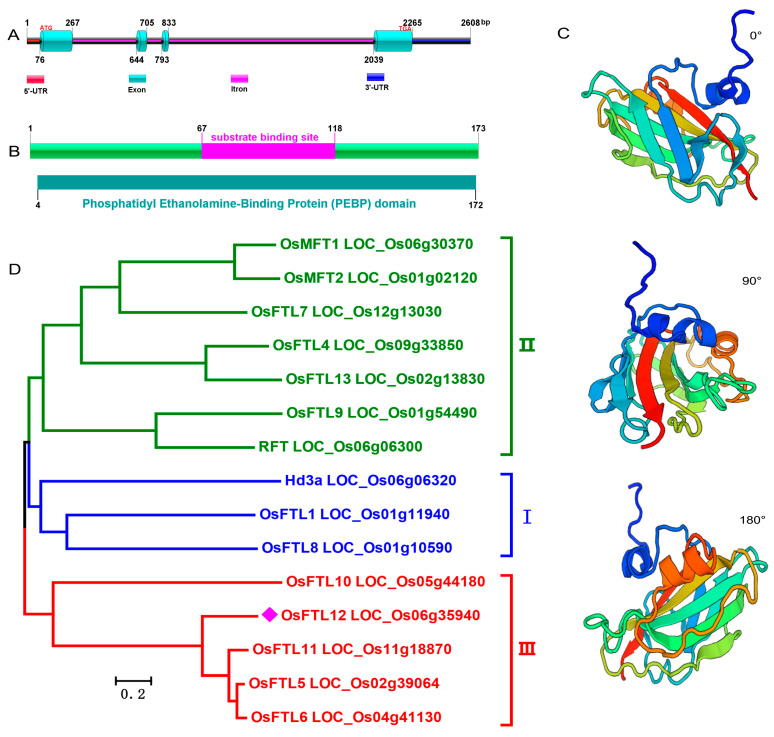
Integrated analysis of the *OsFTL12* DNA and protein sequences. (**A**) schematic diagram of *OsFTL12* genome structure. (**B**) NCBI BlastP predicted the conserved domain in the *OsFTL12* protein. (**C**) SWISS-Model predicted protein 3D structure, with *FT* (PDB 6IGJ) as the template. (**D**) phylogenetic tree of *OsFTL12* paralogues in rice.

**Figure 2 ijms-25-01449-f002:**
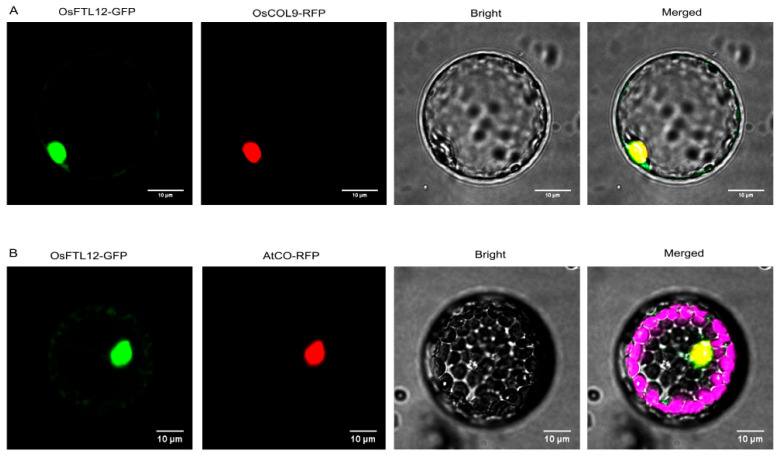
Subcellular localization analysis of *OsFTL12* encoded protein. (**A**) *OsFTL12*-GFP was transiently expressed in rice protoplast cells. The green channel was *OsFTL12*-GFP, the red nucleus marker was *OsCOL9*-RFP, scale bar = 10 µm. (**B**) *OsFTL12*-GFP was co-transformed into the *Arabidopsis* protoplast cell with a nucleus marker. The green channel was *OsFTL12*-GFP, the red channel was *AtCO*-RFP, and the pink Channel was chloroplast auto-fluorescence, scale bar = 10 µm.

**Figure 3 ijms-25-01449-f003:**
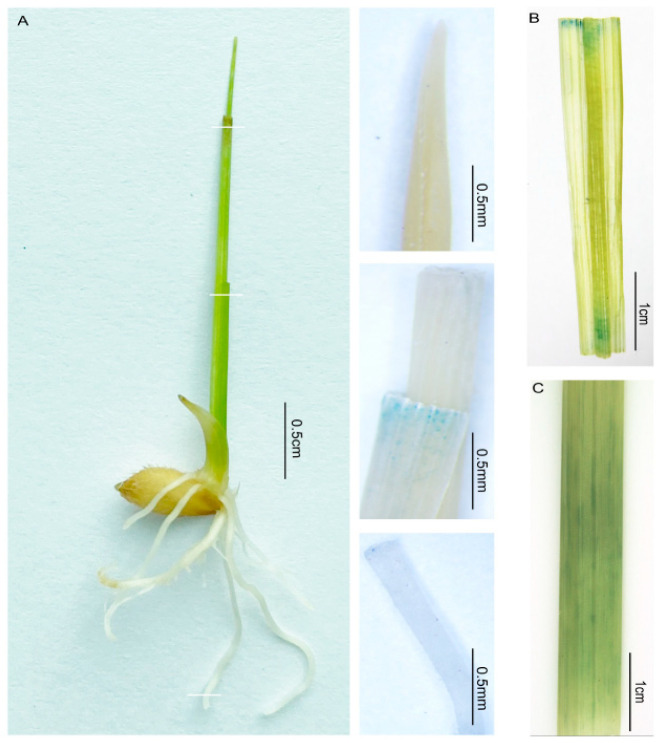
GUS staining identified the activity of the promoter of *OsFTL12*. (**A**) Five-day-old germinated seedling of the P*_OsFTL12_*-GUS transgenic line, leaf tip, GUS stained leaf shoot, and young root. (**B**) The leaf blade of the P*_OsFTL12_*-GUS plant at the booting stage. (**C**) The fresh stem of the P*_OsFTL12_*-GUS plant at the booting stage.

**Figure 4 ijms-25-01449-f004:**
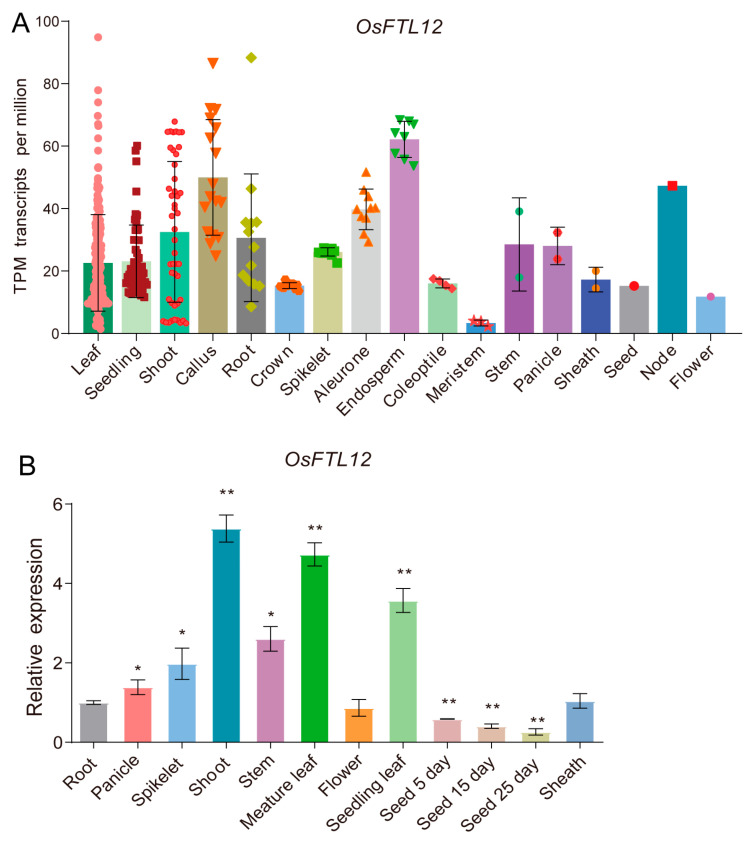
Expression pattern of *OsFTL12* in different tissues. (**A**) *OsFTL12* TPM (Transcripts Per Million) values published in the Rice Expression Database (RED), each dot in the column represents an independent RNA-seq sample. (**B**) qPCR to examine the expression levels of *OsFTL12* in different tissues of wild-type, root was the control sample. All qPCR values shown are the mean ± SD of three independent repeats, an asterisk indicates significant differences when compared with the control group (*t*-test, * *p* < 0.05, ** *p* < 0.01).

**Figure 5 ijms-25-01449-f005:**
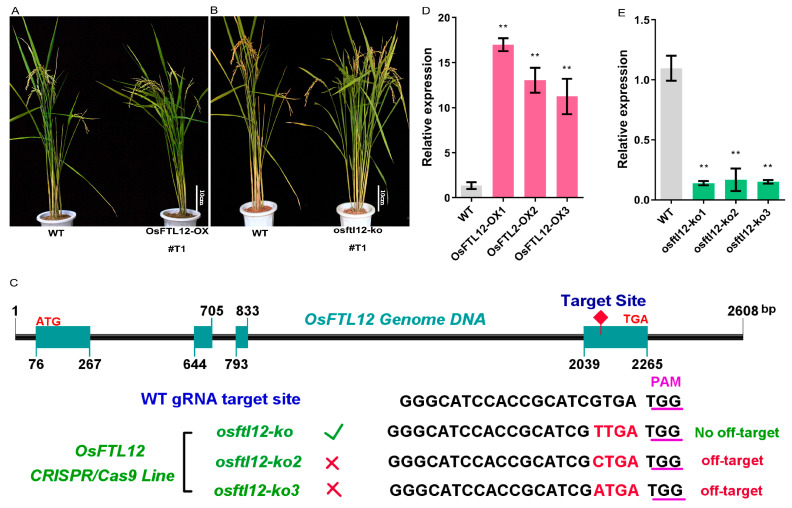
Generation of *OsFTL12* transgenic plants. (**A**,**B**) Phenotypes of WT and *OsFTL12* over-expressing and knock-out transgenic lines (T1 generation), scale bar = 10 cm. (**C**) Nucleotide sequences of mutant CRISPR/Cas9 spliced target sites in the genetically stable *osftl12-ko* lines (T1). Red bases indicate mutant DNA bases. (**D**,**E**) Relative expression levels of *OsFTL12* in transgenic lines (T2 generation). Values shown are means ± SD from three parallel biological replicates. All asterisks indicate significant differences (** *p* < 0.01) compared with the wild type.

**Figure 6 ijms-25-01449-f006:**
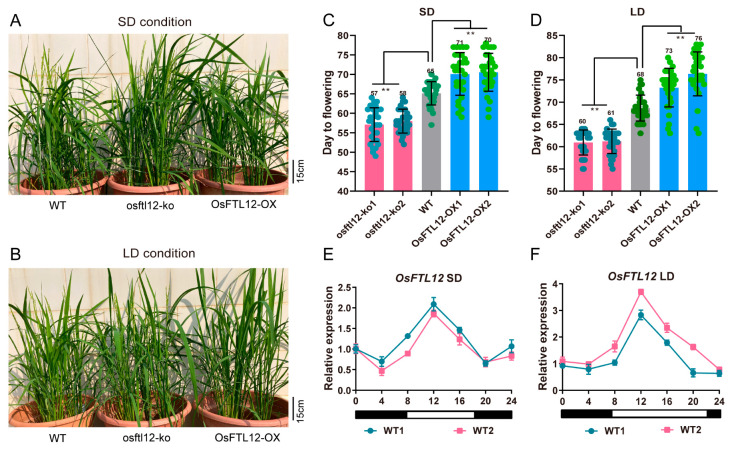
*OsFTL12* mutants promote rice flowering under SD and LD conditions. (**A**,**B**) Phenotypes of WT and *OsFTL12* transgenic lines (T2) under SD and LD conditions. (**C**,**D**) Investigation of heading data in WT and *OsFTL12* transgenic lines (T2) under SD and LD conditions. Values shown are mean ± SD of 30 independent plants (n = 30). Asterisks indicate significant differences (** *p* < 0.01) compared with WT. (**E**,**F**) Circadian expression patterns of *OsFTL12* in WT under SD and LD conditions, respectively. Values shown are mean ± SD of three independent experiments.

**Figure 7 ijms-25-01449-f007:**
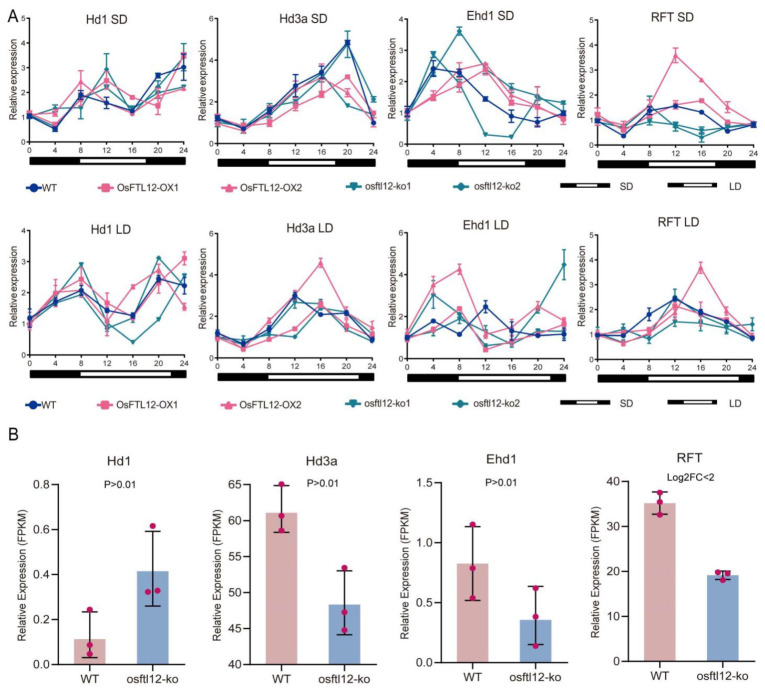
The circadian expression pattern of *Hd1*/*Hd3a* and *Ehd1*/*RFT* in WT and *OsFTL12* transgenic lines (T2) under different day length modes. (**A**) Relative expression levels of *Hd1*/*Hd3a* and *Ehd1*/*RFT* in WT and *OsFTL12* transgenic lines. (**B**) Relative transcript abundances (FPKM) of *Hd1*/*Hd3a* and *Ehd1*/*RFT* in WT and *osftl12-ko* plants by RNA-seq. All values shown are mean ± SD of three independent experiments.

**Figure 8 ijms-25-01449-f008:**
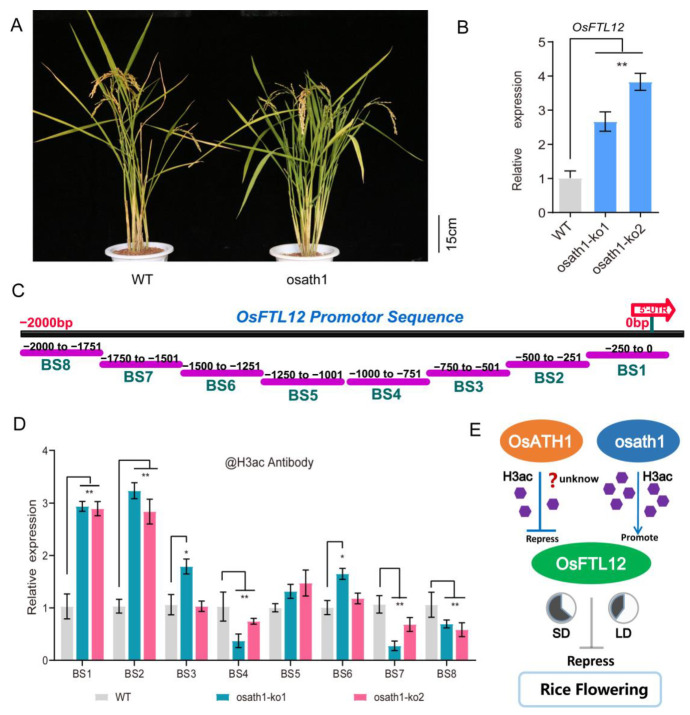
Results of ChIP-qPCR assays on the *OsFTL12* promoter sequence using H3ac antibody in *osath1-ko* plants. (**A**) Photograph of *osath1-ko* plant. (**B**) Relative expression level of *OsFTL12* in *osath1-ko* plants compared with wild-type. (**C**) Schematic structure displaying the ChIP-qPCR detected regions in the *OsFTL12* promoter. (**D**) ChIP-qPCR analysis at *OsFTL12* promoter regions were performed using antibodies against H3ac in *osath1-ko* samples. Values shown are means ± SD from three parallel biological replicates. All asterisks indicate the significant difference (* *p* < 0.05, ** *p* < 0.01) compared with the wild type. (**E**) Putative model of *OsFTL12* modulating the rice flowering under different day-light conditions.

## Data Availability

Data are contained within the article and Appendix A.

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
