# Peer review of "Phosphatidyl Ethanolamine Binding Protein FLOWERING LOCUS T-like 12 (OsFTL12) Regulates the Rice Heading Date under Different Day-Length Conditions"

_ijms, 2024, doi:10.3390/ijms25031449_

Round 1

Reviewer 1 Report

Comments and Suggestions for Authors

In this paper authors defined the role of OsFTL12 in flowering in rice under different day light condition. Which highlighted the OsFTL12 playing crucial role in flowering and had highest expression in stem and leaf. I have one concern, why knockout of OsFTL12 does not affects the leaf and stem features?

Comments on the Quality of English Language

Minor spell check is requires

Author Response

Reviewer1

In this paper authors defined the role of OsFTL12 in flowering in rice under different day light condition. Which highlighted the OsFTL12 playing crucial role in flowering and had highest expression in stem and leaf. I have one concern, why knockout of OsFTL12 does not affects the leaf and stem features?

Response: Thank you very much for your comment. In this study, we identified the molecular function of OsFTL12 in rice flowering. OsFTL12 contained the highest expression levels in rice stem and leaf by GUS-staining and tissue qPCR analysis, but we not found that the OsFTL12 knock-out significantly affected the stem or leaf growth. It is know that the stem and growth depends on the phytohormone pathway, especially the gibberellin (GA), brassinolide (BR), and strigolactone (SL) pathways by modulating the critical genes expression, such as the SD1, OsDWARF4, D14.  However, we found that OsFTL12 knock-out affected the expressions of five GA 2-beta-dioxygenase genes (LOC_Os07g01340, LOC_Os01g55240, LOC_Os04g44150, LOC_Os02g41954, LOC_Os05g43880), and one GA 20 oxidase gene down-regulated expression (LOC_Os03g63970) based on the transcriptome DEGs profile between the wild-type and osftl12-ko plants, and this result not contained the BR and SL related genes. Even though the OsFTL12 affected the expression level of GA biosynthesis genes, and not produced an effect on the BR and SL pathway. But we deduced that antagonism procedure existed in OsFTL12-induced GA genes, such as the OsGA2ox9 (LOC_Os02g41954) up-regulated reduced the stem length, but down-regulated the GNP1 (LOC_Os03g63970) promoted the GA content increasing, therefore the GA synthesis genes not affected the stem or leaf growth. On the other hand, OsFTL12 expression level shown higher in stem and leaf, not represented that the OsFTL12 protein content had the similar tendency to regulate the stem and leaf growth. Finally, we also speculated that OsFTL12 lost function, maybe another rice FTL homologous genes replaced the OsFTL12 deficient to maintain the normal plant growth. We have deposited this reply about the OsFTL12 not affected the stem or leaf growth into the Discussion.

Totally, your good suggestion provided a scientific problem for our next exploring the OsFTL12 function on the regulation of stem and leaf growth.

Reviewer 2 Report

Comments and Suggestions for Authors

In this study, the intricate roles of Plant FLOWERING LOCUS T-Like (FTL) genes, particularly OsFTL12 in rice reproductive stages, were analyzed. The OsFTL12 protein, housing the PEBP domain and localizing in the nucleus, displayed specific transcript expression in the shoot and leaf blade with high abundance. Despite a discrepancy in GUS staining results, the OsFTL12 promoter exhibited heightened activity in the leaf and stem. Knocking out OsFTL12 resulted in an early flowering phenotype under various day conditions, independent of Hd1/Hd3a and Ehd1/RFT pathways, establishing OsFTL12 as a key regulator of flowering. Moreover, the study identified OsATH1 as an upstream regulator affecting OsFTL12 expression through Histone H3 acetylation, offering potential breeding strategies for manipulating rice early heading behavior. The manuscript is interesting and valuable, its novelty is clear. I have some minor technical suggestions regarding the text.

-          Keywords should be arranged alphabetically and not repeating words from the title.

-          Line 38: Angiosperms is not a kingdom.

-          The name of the cultivar should not be written in italics but in inverted comas (‘’).

-          From where did you obtain the plant material?

-          What was the PPFD in the growth chamber?

-          Line 158: reference is missing.

-          Producers of key chemicals and equipment are missing.

-          Unit style is incorrect (it should be mg·L-1).

-          A statistical analysis chapter is missing in the Materials and methods.

-          Line 390: space bar error.

-          I suggest preparing a separate Conclusions chapter.

-          Please follow the MDPI formatting style in the Reference list.

-          DOI numbers are missing.

After incorporating all the necessary changes, the manuscript can be accepted for publication.

Comments on the Quality of English Language

English is fine.

Author Response

Reviewer2

In this study, the intricate roles of Plant FLOWERING LOCUS T-Like (FTL) genes, particularly OsFTL12 in rice reproductive stages, were analyzed. The OsFTL12 protein, housing the PEBP domain and localizing in the nucleus, displayed specific transcript expression in the shoot and leaf blade with high abundance. Despite a discrepancy in GUS staining results, the OsFTL12 promoter exhibited heightened activity in the leaf and stem. Knocking out OsFTL12 resulted in an early flowering phenotype under various day conditions, independent of Hd1/Hd3a and Ehd1/RFT pathways, establishing OsFTL12 as a key regulator of flowering. Moreover, the study identified OsATH1 as an upstream regulator affecting OsFTL12 expression through Histone H3 acetylation, offering potential breeding strategies for manipulating rice early heading behavior. The manuscript is interesting and valuable, its novelty is clear. I have some minor technical suggestions regarding the text.

Keywords should be arranged alphabetically and not repeating words from the title.

Response: Thank you for your suggestion. We have revised the Keywords by following your suggestion. The new Keywords: AT-hook protein, Flowering time, FTL homologous, H3 acetylation, Transcription regulation.

Line 38: Angiosperms is not a kingdom.

Response: Thank you for your comment. We have realized our mistake, we have corrected this error. Angiosperms kingdom has been replaced by the flowering plants (angiosperms).

The name of the cultivar should not be written in italics but in inverted comas (‘’).

Response: Thank you for your comment, we have revised this error.

From where did you obtain the plant material?

Response: Plant material was taken from the ‘Nipponbare’ cultivar of the rice plant Oryzae sativa Geng, this was used as the wild-type plant in this study. The ‘Nipponbare’ seed was collected and stored as a rice germplasm in the College of Coastal Agricultural Sciences, Guangdong Ocean University. The ‘Nipponbare’ seed was collected by the Prof. Jianfu Guo who saved this plant material in 2006 at the College of Coastal Agricultural Sciences, Guangdong Ocean University.

What was the PPFD in the growth chamber?

Response: Light intensity for plant growth is typically measured as photosynthetic photon flux density (PPFD), the chamber PPFD value was 500 µmol·m2·s-1, this parameter is eminently suit for plant growth.

Line 158: reference is missing.

Response: Thank you very much for your suggestion. We have listed all of the references in the section of 4.2. Bioinformatics analysis

EvolView: Zilong He, Huangkai Zhang, Shenghan Gao, Martin J Lercher, Wei-Hua Chen, Songnian Hu. Evolview v2: an online visualization and management tool for customized and annotated phylogenetic trees. Nucleic Acids Res. 2016 Jul 8;44(W1):W236-41.

IBS 1.0 version software: Liu W, Xie Y, Ma J, Luo X, Nie P, Zuo Z, Lahrmann U, Zhao Q, Zheng Y, Zhao Y, Xue Y, Ren J. IBS: an illustrator for the presentation and visualization of biological sequences. Bioinformatics. 2015 Jun 10. pii: btv362.

SWISS-modeling: Waterhouse, A., Bertoni, M., Bienert, S., Studer, G., Tauriello, G., Gumienny, R., Heer, F.T., de Beer, T.A.P., Rempfer, C., Bordoli, L., Lepore, R., Schwede, T. SWISS-MODEL: homology modelling of protein structures and complexes. Nucleic Acids Res. 46(W1), W296-W303 (2018).

The Rice Expression Database (RED): Xia L#, Zou D#, Sang J, Xu XJ, Yin HY, Li MW, Wu SY, Hu SN, Hao LL*, Zhang Z*: Rice Expression Database (RED): an integrated RNA-Seq-derived gene expression database for rice. Journal of Genetics and Genomics, 2017, 44(5):235-241.

PLACE predicted the cis-elements: K Higo, Y Ugawa, M Iwamoto, H Higo. PLACE: a database of plant cis-acting regulatory DNA elements. Nucleic Acids Res

. 1998 Jan 1;26(1):358-9.

The Plant regulomics database: Xiaojuan Ran#, Fei Zhao#, Yuejun Wang#, Jian Liu, Yili Zhuang, Luhuan Ye, Meifang Qi, Jingfei Cheng and Yijing Zhang*. (2019), Plant Regulomics: A Data-driven Interface for Retrieving Upstream Regulators from Plant Multi-omics Data[J]. The Plant Journal. doi: 10.1111/tpj.14526

Producers of key chemicals and equipment are missing.

Response: Thank you very much for your suggestion, we have provided all of the key chemical and equipment producer information.

Unit style is incorrect (it should be mg·L-1).

Response: Thank you very much for your suggestion, we have corrected this error.

A statistical analysis chapter is missing in the Materials and methods.

Response: Thank you very much for your suggestion, we have provided the statistical analysis chapter. 4.9. Statistical analysis

OsFTL12 expression level in different tissue was caculated the real-time PCR value with three independent repeats, the histogram represented the mean ± SD, and significance analysis was carried out with t-test (*P<0.05, **P<0.01). OsFTL12 expression examination in transcgenic plants followed the above-mentioned statistical method. The plant materials heading date was recorded as the day when the panicle grew out from the leaf sheath with 5 cm. 30 independent plants (n=30) were caculated the flowering time, the histogram represented the mean ± SD, and significance analysis was carried out with t-test (*P<0.05, **P<0.01). Moreover, rice leaf blade was harvested at four hour intervals and extracted the total mRNA to examine the expression levels of Hd1, Hd3a, RFT, Ehd1 under SD and LD, the line chart indicated their expression levels with mean ± SD in three independent repeats. All the statistical analysis figure was employed the GraphPad Prism 9.

Line 390: space bar error.

Response: Thank you very much for your suggestion, we have corrected this error.

I suggest preparing a separate Conclusions chapter.

Response: Thank you very much for your suggestion. We have provided a separate Conclusion.

OsFTL12 encodes a nucleus localized proten with a conserved PEBP domain, which  highly expresses in the shoot and leaf and over-expression delayes the rice flowering time under SD and LD growth condition. Additionally, OsATH1 acts as a transcription factor at the upstream of OsFTL12 promoter to regulated the OsFTL12 expression by H3 acetylation pathway. Our study contributs a theoretical basis for future breeding the early-maturing variety by manipulating the expression of OsFTL12 in rice.

Please follow the MDPI formatting style in the Reference list. DOI numbers are missing.

Response: Thank you very much for your suggestion, we have corrected this error.

After incorporating all the necessary changes, the manuscript can be accepted for publication

Response: Thank you very much for your suggestion.

Reviewer 3 Report

Comments and Suggestions for Authors

Manuscript ijms-2808543 “Phosphatidyl Ethanolamine Binding Protein FLOWERING LOCUS T-Like 12 (OsFTL12) Regulates the Rice Heading Date Under Different Day-length Conditions” by Huang et al. presents an interesting study about gene expression analysis of OsFTL12 in rice and the regulation of heading date.

This manuscript presents a valuable study with production and breeding applications. However, this manuscript is very confused in some parts mainly in the description of the methodology and the experimental design. In addition, the manuscript presents important deficiencies mainly in the discussion of results. Phenotype results must be clarified. Validation of RNA-Seq through qPCR is not clear. For these reasons, this manuscript is ACCEPTABLE for publication in the International Journal of Molecular Sciences after a major revision.

The major points for the REVISION of the manuscript are:

Around the whole manuscript, the complete name of genes should be in italics.

 Objectives are very large. Authors must simplify the objectives not including any methodological reference.

 Plant material must be completed describing main agronomic traits of the rice genotype assayed.

 Phenotyping must be clarified in the Methodology section. Evaluation of the heading date and the evaluation of the growth of the rice must be clarified in the description of growth conditions.

 From the methodology point of view, RNA-Seq and differential expression analysis are the key methodology used. RNA-Seq characteristics must be completed in section 2.7. In addition, biological and technical replicates must be described. Finally the origin of the tissue analysed in both assays (qPCR and RNA-Seq) must be clarified, flower, fruits, leaves, etc.

 qRT-PCR analysis should be also clarified indicating the nature of the assayed technical and biological samples. In my opinion RNA-Samples must come from a different assay, not exactly the same that the RNA-Seq assay. This question must be clarified. In addition, the selection of the candidate genes must be justified. If it is possible a higher number of genes to validate RNA-Seq data should increase the robustness of the experiment.

 Description of Results is very poor, mainly regarding phenotype analysis. The description of rice genotype and their response is deficient.

 A new table with the response of the assayed genotype must be included in the results section completing data from Figures 2 and 3.

 Quality of Figure 7 must be revised increasing font size.

 In Figure 7 authors should be incorporated the correlation coefficient between qPCR and RNA-Seq data in the assayed genes.

 Discussion section is very week. Authors must clarify the novelty of the obtained results in comparison with previous transcriptome data. It is necessary to transform RNA-Seq data in biological data, this is the nature of this manuscript. However, this biological data should be discussed in term of biological information adding some biological hypothesis clarifying the development of the peach fruit shape. The election of some genes for the monitoring of this process is also very important in the Discussion section.

 A new Conclusion section must be incorporated. Authors must indicate the main implications of these results from an agronomical and breeding point of view.

Comments on the Quality of English Language

English is OK

Author Response

Reviewer3:

Manuscript ijms-2808543 “Phosphatidyl Ethanolamine Binding Protein FLOWERING LOCUS T-Like 12 (OsFTL12) Regulates the Rice Heading Date Under Different Day-length Conditions” by Huang et al. presents an interesting study about gene expression analysis of OsFTL12 in rice and the regulation of heading date.

This manuscript presents a valuable study with production and breeding applications. However, this manuscript is very confused in some parts mainly in the description of the methodology and the experimental design. In addition, the manuscript presents important deficiencies mainly in the discussion of results. Phenotype results must be clarified. Validation of RNA-Seq through qPCR is not clear. For these reasons, this manuscript is ACCEPTABLE for publication in the International Journal of Molecular Sciences after a major revision.

Response: thank you very much for your comments, we have improved the quality of the manuscript based on your suggestion, the revised sections have been listed in the track version, please check it.

The major points for the REVISION of the manuscript are:

  1. Around the whole manuscript, the complete name of genes should be in italics.

Response: thank you for your suggestion, the total of genes names have been corrected with italics.

  1. Objectives are very large. Authors must simplify the objectives not including any methodological reference.

Response: thank you very much for your suggestion, we have re-written the objectives in the improved version. Last section of Introduction.

Plant material must be completed describing main agronomic traits of the rice genotype assayed.

Response: Plant material was taken from the ‘Nipponbare’ cultivar of the rice plant oryza sativa L. japonica, this was used as the wild-type plant in this study. The ‘'Nipponbare'’ seed was collected and stored as a rice germplasm by Prof. Jianfu Guo in the College of Coastal Agricultural Sciences, Guangdong Ocean University. ‘Nipponbare’ cultivar genome information (Rice Genome Annotation Project, RGAP) is completely released earlier than another rice varieties, and this material has been broadly used in rice study.

Phenotyping must be clarified in the Methodology section. Evaluation of the heading date and the evaluation of the growth of the rice must be clarified in the description of growth conditions.

Response: thank you very much for your suggestion, we have provided the phenotype assay in the methodology section.

Plant material was taken from the 'Nipponbare' cultivar of the rice plant Oryza Sativa L. japonica (Geng in China), this was used as the wild-type plant in this study. The 'Nipponbare' seed was collected and stored as a rice germplasm by Prof. Jianfu Guo in the College of Coastal Agricultural Sciences, Guangdong Ocean University. Wild-type and transgenic plants were cultured in a constant temperature (28 ℃ daylight, 25 ℃ dark) and humidity incubator. The plants were cultured on short (SD, 10 hours daylight, 14 hours dark) and long day (LD, 14 hours daylight, 10 hours dark) light cycles respectively. Light intensity for plant growth is typically measured as photosynthetic photon flux density (PPFD), the chamber PPFD value was 500 µmol·m2·s-1, this parameter is eminently suit for plant growth. The seeds of all plant materials germinated from the soil, this day was recorded as the first day. The heading date was recorded as the day when the rice panicle grew out from the leaf sheath with 5 cm, and 30 independent plants were recorded their heading data. When the different rice materials (wild-type and transcgenic plants) flowering rate was 90%, all the plant materials were removed from the chamber to a greenhouse until harvested their seeds. After being cultured for 40 days in the chamber under SD and LD condition, the rice leaf blade was harvested at four hour intervals, mRNA was then extracted to detect the circadian expression passion of the relevant genes.

From the methodology point of view, RNA-Seq and differential expression analysis are the key methodology used. RNA-Seq characteristics must be completed in section 2.7. In addition, biological and technical replicates must be described. Finally the origin of the tissue analysed in both assays (qPCR and RNA-Seq) must be clarified, flower, fruits, leaves, etc.

Response: thank you very much for your suggestion, we have provided a detail description regrading of RNA-seq experiment and analysis in the section of 2.7.7 RNA-seq. Total RNA was extracted from 45 day-old wild-type and osftl12-ko leaves (mixed three blades, liquid nitrogen ground 100 mg leaf powder), using the reagent TRIzol (Cat#15596026, Invitrogen, USA). Each sample (wild-type and osftl12-ko) had three independent biological repeats with six independent RNA-seq libraries.

qRT-PCR analysis should be also clarified indicating the nature of the assayed technical and biological samples. In my opinion RNA-Samples must come from a different assay, not exactly the same that the RNA-Seq assay. This question must be clarified. In addition, the selection of the candidate genes must be justified. If it is possible a higher number of genes to validate RNA-Seq data should increase the robustness of the experiment. 

Response: thank you very much for your suggestion. The qRT-PCR of flowering genes expression (OsFTL12, Hd1, Hd3a, Ehd1, RFT) was carried out with mRNA extraction in 45 old-day rice materials, and the RNA-seq experiment was carried out in the 45 old-day rice materials (WT, osftl12-ko). The selected candidate genes have been validated in the Figure 7, and we have revised the qRT-PCR methodology section.

RNA extraction and qPCR followed our previous published study [25]. For flowering genes expression (OsFTL12, Hd1, Hd3a, Ehd1, RFT) examination, the rice materials were firstly cultured for 45 days in the chamber under SD and LD condition, the rice leaf blade was harvested at four hour intervals, mRNA was then extracted to detect the circadian expression patterns of the relevant genes. total RNA was extracted from 100 mg of 45 day-old wild-type and transgenic (OsFTl12-OX, osftl12-ko, osath1-ko) plants leaf using Trizol Reagent (Cat#15596026, Invitrogen, USA), then reverse-transcribed using a PrimeScript RT reagent Kit (Cat#FSK-100, TOYOBO, Janpan). The 100 ng cDNA was quantified in a 20 µl reaction volume with SYBR Premix ExTaq™ (Cat#RR390A, TaKaRa, Janpan) by using the ABI StepOne Plus system (Cat#4376600, Applied Biosystems, USA). The rice actin gene was used as an internal reference. The relative Gene expression levels as a fold change compared to control sample (wild-type) were calculated using the 2-ΔΔCT calculating method with three biological repeats[55]. For OsFTL12 expression in different tissues followed the same method, the total mRNA was extracted from the one week-old wild-type (Nipponbare) seedling root, sheath, leaf; 45 day-old wild-type shoot, stem, mature leaf; 65 day-old wild-type panicle, spikelet, flower; and the seed developed at 5, 15, 25 day during the period of grain-filling.

Description of Results is very poor, mainly regarding phenotype analysis. The description of rice genotype and their response is deficient.

Response: thank you very much for your suggestion. We have improved the Results section, deleted more redundant description, and provided a description of rice phenotype in the heading stage. 2.6. OsFTL12 over-expression delays heading date in rice

A new table with the response of the assayed genotype must be included in the results section completing data from Figures 2 and 3.

Response: Four-weeks old Arabidopsis thaliana seedling leaves (ecotype Columbia) and two-weeks old rice seedling (wild-type plant, Oryza Sativa L. japonica cv. Nipponbare) sheaths in Figure 2. The 'Nipponbare' was used as the wild-type plant for the transgenic experiment for GUS-staining in Figure 3. We have provided the description in the Methodology section.

Quality of Figure 7 must be revised increasing font size.

Response: Thank you very much for your suggestion, we have re-created the Figure 7, the improved Figure 7 contained a good font size.

In Figure 7 authors should be incorporated the correlation coefficient between qPCR and RNA-Seq data in the assayed genes.

Response: Thank you very much for your suggestion, we have re-created the Figure 7, the RNA-seq result of Hd1, Ehd1, Hd3a, RFT have been displayed in the Figure 7.

Discussion section is very week. Authors must clarify the novelty of the obtained results in comparison with previous transcriptome data. It is necessary to transform RNA-Seq data in biological data, this is the nature of this manuscript. However, this biological data should be discussed in term of biological information adding some biological hypothesis clarifying the development of the peach fruit shape. The election of some genes for the monitoring of this process is also very important in the Discussion section.

Response: Thank you very much for your suggestion, we have described the RNA-seq process in the section of methodology, the RNA-seq contained three independent biological repeats. On the other hand, the we have revised the Discussion.

A new Conclusion section must be incorporated. Authors must indicate the main implications of these results from an agronomical and breeding point of view.

Response: Thank you very much for your suggestion, we have provided the conclusion.

  1. Conclusion

OsFTL12 encodes a nucleus localized proten with a conserved PEBP domain, which  highly expresses in the shoot and leaf and over-expression delayes the rice flowering time under SD and LD growth condition. Additionally, OsATH1 acts as a transcription factor at the upstream of OsFTL12 promoter to regulated the OsFTL12 expression by H3 acetylation pathway. Our study contributs a theoretical basis for future breeding the early-maturing variety by manipulating the expression of OsFTL12 in rice.

Round 2

Reviewer 3 Report

Comments and Suggestions for Authors

Authors have greatly improved the revised manuscript following the comments of the reviewer. This revised manuscript provides a meaningful knowledge for the molecular characterization of heading date in rice. For this reason, I recommend accepting this manuscript.